# Whole-genome sequencing of *Hyphopichia burtonii* from isolated yeast recovered from zebra dove droppings in Thailand

Saowakon Indoung[1☯], Sanicha Chumtong[2☯], Sakaoporn Prachantasena[1],
Ratchakul Wiriyaprom[1], Komwit Surachat[2,4], Rattanaruji Pomwised[3],
Ruttayaporn Ngasaman[1]*

1 Faculty of Veterinary Science, Prince of Songkla University, Songkhla, Thailand, 2 Department of Biomedical Sciences and Biomedical Engineering, Faculty of Medicine, Prince of Songkla University, Songkhla, Thailand, 3 Division of Biological Science, Faculty of Science, Prince of Songkla University, Songkhla, Thailand, 4 Translational Medicine Research Center, Faculty of Medicine, Prince of Songkla University, Songkhla, Thailand

☯ These authors contributed equally to this work
*ruttayaporn.n@psu.ac.th

## Abstract

This study aimed to characterize zoonotic yeasts from zebra dove (*Geopelia striata*) droppings in small farms in Songkhla province, Thailand. Four out of thirty-one isolates were found with morphology and biochemical test results like those of *Cryptococcus* spp. (12.9%) but they exhibited different results from the positive control in nested polymerase chain reaction (PCR); the first step with ITS1-ITS4 primers was negative, but the nested PCR step with CN4-CN5 was positive. All isolates were subjected to antifungal susceptibility testing. Only the Tip11 isolate showed high resistance to itraconazole and ketoconazole, with minimum inhibitory concentrations of >32.0 and 32.0 μg/mL, respectively, corresponding to high levels of minimum fungicidal concentrations for both drugs (>32 μg/mL). Consequently, Tip11 was selected as a representative of the contaminated isolates and subjected to whole-genome sequencing analysis. The results identified the Tip11 isolate as *Hyphopichia burtonii* (99.30% identity), with a genome length of 12,360,159 bp, a GC content of 35.16%, and 2,146 protein groups closely related to the *Saccharomycetaceae* family. It showed 43 candidates resistance genes of antifungal drugs mostly in azole and itraconazole group which related to the antifungal susceptibility testing. This is the first report of *H. burtonii* in zebra doves in Thailand. This yeast has been previously identified as a cause of human infection leading to peritonitis in Thailand, and its resistance to antifungal drugs may pose a public health risk. Therefore, the application of biosecurity measures on farms, such as regular removal of droppings and cage sanitization, should be implemented according to good agricultural practices.

**Data availability statement:** All relevant data are within the paper and its Supporting Information files.

**Funding:** Funding from Faculty of Veterinary Science, Prince of Songkla University Grant no. VET6604128S The funders had no role in study design, data collection and analysis, decision to publish, or preparation of the manuscript.

**Competing interests:** The authors have declared that no competing of interests.

## Introduction

The zebra dove (*Geopelia striata*) is renowned for its soft, staccato cooing calls, particularly in southern Thailand. These birds are popular as pets, especially for cooing competitions, making the breeding of zebra doves a thriving industry focused on producing birds with desirable vocal qualities. Many farmers rear zebra doves for sale and export to countries such as Indonesia, Malaysia, and Brunei. In Songkhla province, particularly in the Chana district, there are more than 170 zebra dove farms. The are large, moderate and small farms 30, 40, and 100, respectively. Collectively, approximately 40,000–50,000 zebra dove in the area [1]. From observation, individuals who raise zebra doves maintain close contact with the birds because the cages are often located inside or near their homes. Champion birds are usually kept as close to their owners as possible.

Various yeasts can infect pet birds, including *Candida albicans*, *Aspergillus* spp., *Cryptococcus neoformans*, and *Trichomonas gallinae* [2–4]. Avian gastric yeast (*Macrorhabdus ornithogaster*), the causative agent of proventriculitis in birds, is also found in these settings. Clinical signs of yeast infection can range from acute symptoms, resulting in sudden death, to chronic symptoms such as wasting, diarrhoea, or enteritis [5]. In psittacines, pathogenic yeasts have been isolated from faecal or cloacal samples in passeriformes, columbiformes, and falconiformes [6,7]. In one study, *Candida catenulata* and *Candida albicans* were the most frequently isolated species from cloacal swabs, faeces, and eggs of laying hens [8].

Birds harbour potentially pathogenic yeasts and can easily spread them into the environment via their faeces. Examples of zoonotic yeasts associated with bird droppings include *Candida kefyr*, *Cryptococcus neoformans*, and *Cryptococcus gattii* [9,10]. However, yeasts in the genus *Hyphopichia* have not been previously reported in zebra doves. Even rare cases have been documented in humans, but it is considered as a zoonotic yeast. This study was performed to analyze the assembly sequence of *H. burtonii* yeast obtained from whole-genome sequencing (WGS). The results may enhance understanding and raise awareness regarding this yeast.

## Materials and methods

### Fungal isolation

Four contaminated isolates, initially suspected to be zoonotic yeast (*Cryptococcus* spp.) were recovered from zebra dove droppings by Indoung et al [3]. The isolates were stored at −80°C in glycerol stock tubes and sub-cultured onto Sabouraud dextrose agar (SDA) (Oxoid, UK). Staining with India ink and crystal violet was performed to identify capsules and yeast budding. Candida albicans ATCC 90028 was used for quality control. The inoculum was prepared from 48-hour-old cultures suspended in 0.85% normal saline to a concentration of 0.5 McFarland ($1–5 \times 10^6$ CFU/mL) using a spectrophotometer (absorbance = 530 nm) and diluted to $0.5–2.5 \times 10^3$ CFU/mL in RPMI 1640 broth medium [11].

## Yeast DNA extraction

Genomic DNA was prepared for nested polymerase chain reaction (PCR) and WGS. The cryopreserved cells in 20% (v/v) glycerol were streaked onto SDA (HiMedia, India) and incubated at 37°C for 24 hours. Single fungal colonies were then cultured on SDA at 37°C for 48 hours. Genomic DNA was extracted from all single fungal colonies using the Presto™ Mini gDNA Yeast Kit (Geneaid Biotech, Taiwan) in accordance with the manufacturer's instructions. The genomic DNA was analysed using 1% agarose gel electrophoresis, with quality and quantity measured to have an A260/280 ratio of 1.8–2.0 and a concentration of at least 50 ng/μL, determined by a NanoDrop™ 2000 spectrophotometer (Thermo Fisher Scientific, USA).

## Nested PCR targeting *Cryptococcus* spp

To rule out *Cryptococcus* spp., pure DNA was subjected to modified nested PCR as previously reported by Mitchell [12]. In the first step of amplification, the primers ITS-1 (5′-TCCGTAGGTGAACCTGCGG-3′) and ITS-4 (5′-TCCTCCGCTTATT GATATGC-3′) were used, resulting in an amplicon of approximately 500–600 bp. The 12-μL PCR mixture contained 10 μL of KAPA PCR Ready Mix (Kapa Biosystems, Japan), 0.5 μL of 10 μM of each primer, and 1 μL of DNA template. The amplification conditions were as follows: heating at 94°C for 5 minutes, followed by 35 cycles of denaturation at 94°C for 30 seconds, annealing at 55°C for 45 seconds, and DNA extension at 72°C for 1 minute, with a final heating at 72°C for 7 minutes in a Bio-Rad Thermal Cycler (Bio-Rad Laboratories Inc., USA). The second reaction was prepared similarly to the first round, using the amplicon product from the first round as the template, with the primers CN-4 (5′-ATCACCTTCCCAC TATT CACACATT-3′) and CN-5 (5′-GAAGGGCATGCCTGTTT GAGAG-3′), producing an amplicon of 136 bp. The reaction conditions were the same as those used in the first round.

## Drug susceptibility

Testing was performed using the broth microdilution method, following the guidelines in Clinical and Laboratory Standards Institute (CLSI) document M27-A3 [11] with some modifications. RPMI 1640 medium, containing glutamine and lacking bicarbonate (Gibco, USA), was buffered to a pH of 7.0 using 0.165 mol/L 3-(N-morpholino) propanesulfonic acid (MOPS; Sigma, USA) and supplemented with 2% glucose. The medium was filter-sterilised by passing it through a 0.22-μm-pore filter before use in susceptibility tests. Stock solutions of the antifungal powder amphotericin B (PanReac, Spain) and itraconazole (Sigma, USA) were prepared in dimethyl sulfoxide (Amresco Inc, USA), while stock solutions of fluconazole (Sigma, USA) were prepared in water. The antifungal drugs amphotericin B, fluconazole, ketoconazole, and itraconazole were tested at final concentrations of 0.0625–32.00, 0.0125–64.00, 0.0625–32.00, and 0.0625–32.00 μg/mL, respectively. Each well was filled with 100 μL of each antifungal dose, followed by 100 μL of the yeast inoculum. For controls, 100 μL of RPMI 1640 broth medium without antifungal was used as the medium control, and 100 μL of RPMI 1640 broth medium with yeast inoculum was used as the growth control. The test was carried out in duplicate. Plates were incubated at 35°C±2°C for 48 hours, after which 0.01% resazurin was added and the plates were re-incubated for 3 hours to measure viable cells [13]. The minimum inhibitory concentration (MIC) was visually determined by the change in colour from pink/red to blue after 72 hours of incubation. MIC endpoints were read according to NCCLS criteria (previous name of CLSI) [14]. Wells showing no visible growth were subsequently cultured on SDA and incubated at 35°C±2°C for 24 hours to determine the minimum fungicidal concentration (MFC).

## WGS genome assembly and genome annotation

Genomic DNA was subjected to short-read WGS using the MGISEQ-2000 platform (MGI, China) with 150-bp paired-end reads. The raw reads (FASTQ format) were de novo assembled with the sequencing data available at NCBI BioProject ID PRJNA1047299 and BioSample ID SAMN38529121 by using SPAdes version 3.12 to generate genome contigs and

scaffolds [15]. The complete assembled sequences (FASTA format) were verified using the BUSCO version of Galaxy 5.5.0 + galaxy0 (https://shorturl.at/wdpvm) and annotated using the Augustus program version of + galaxy 11.12 (https://shorturl.at/noINI). Protein analysis was conducted by using BUSCO program version of Galaxy version 5.5.0 + galaxy0 (https://shorturl.at/wdpvm) and online OrthoDB database (https://www.orthodb.org/). Both statistics of assembled and annotated sequences were computed by QUAST v.5.0.2 [16].

### Pairwise average nucleotide identity (ANI) and phylogenetic analyses

The 24 published genomes were isolated from 2 geniuses from *Cryptococcus* sp (n = 11). and *Hyphopichia* sp (n = 13). These genomes were available at the National Center for Biotechnology Information (NCBI) under Submitted GenBank assembly number GCA_001661395.1, GCA_003856795.1, GCA_018343885.1, GCA_030573595.1, GCA_037044105.1, GCA_003856775.1, GCA_001599095.1, GCA_030444945.1, GCA_030578475.1, GCA_030569195.1, GCA_030556945.1, GCA_030560785.1, GCA_030569855.1, GCA_001720195.2, GCA_000836335.2, GCA_036417295.1, GCA_002954075.1, GCA_000091045.1, GCA_000185945.1, GCA_000835755.2, GCA_001720205.1, GCA_001720155.1, GCA_000149245.3 and GCA_000149385.1. All previously published *Cryptococcus* sp and *Hyphopichia* sp genomes from NCBI and one of *Hyphopichia* sp. of this study were used Augustus program for annotation by Galaxy version 3.4.0 + galaxy 11.12 (https://shorturl.at/noINI). Then, the annotated sequences (FASTA format) of protein were constructed phylogenetic trees to compare with all annotated protein sequences by OrthoFider version 2.5.5 [17] and using geneious tree builder of Geneious Prime program with Neighbor-joining (1,000 bootstrap). A total of 24 From all the assembled sequences (FASTA format) from NCBI database and one of assembled sequences of this study. All FASTA files (n = 25) were comparative pairwise average nucleotide identity (ANI). The ANI was evaluated using FastANI v1.32 [18].

## Results

Four contaminated isolates, initially presumed to be *Cryptococcus* spp. based on their morphology and the results of biochemical tests by Indoung [3], were recovered on SDA for 3 days at 35°C. Colonies appeared as round cells with elevated and raised surfaces. The margins of the colonies were smooth but irregular and exhibited an opaque, creamy-white colour but not mucoid like colony of *Cryptococcus* spp. Staining the colonies with India ink and Gram stain revealed yeast capsules and budding like those seen in *Cryptococcus* spp. (Fig 1). Similarity to *Cryptococcus* species, the results of the biochemical tests revealed that the urease test was positive, assimilation of glucose, maltose, raffinose and sucrose were

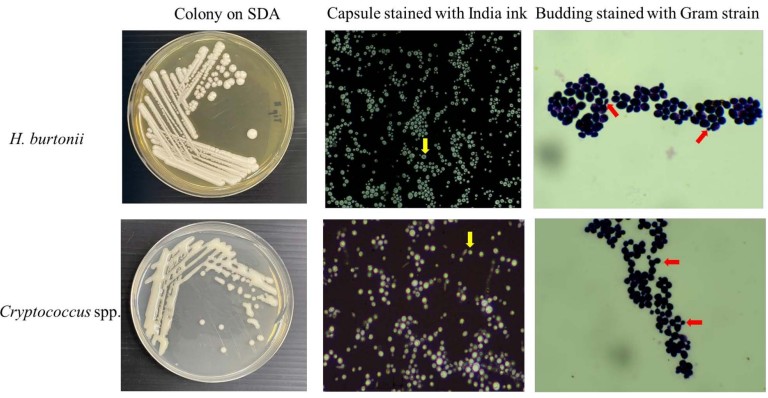

**Fig 1. Comparison the morphology of *H.burtonii* with *Cryptococcus* spp. grown for 3 days on SDA.** Capsule stained with India ink (yellow arrow) and Budding yeast stained with Gram stain (red arrow).

all positive whereas lactose negative. The isolates were then subjected to nested PCR to confirm or rule out *Cryptococcus* spp. All isolates yielded different results from the positive *Cryptococcus* spp. controls, showing negative results in the first step and positive results in the second step of nested PCR (Fig 2).

Antifungal susceptibility testing demonstrated the MIC and MFC values of amphotericin B, fluconazole, itraconazole, and ketoconazole for the four isolates. A low MIC value for amphotericin B was observed in all isolates (0.0625–0.25 µg/mL). By contrast, all isolates exhibited high MIC values for fluconazole (4–8 µg/mL). The MIC value of ketoconazole for Tip1, Tip10, and Park4 was low (0.0625 µg/mL), but for Tip11, it was high (32 µg/mL). The MFC values of all isolates were high for fluconazole (32–64 µg/mL), itraconazole (4.0–32.0 µg/mL), and ketoconazole (4.0–32.0 µg/mL), while the MFC values for amphotericin B ranged from 0.25 to 1.0 µg/mL (Table 1).

Based on the results of morphology, biochemical tests, nested PCR, and antifungal susceptibility testing, Tip11 was selected for WGS. The quality assessment results for WGS using the raw reads (FASTQ format) with the BUSCO program showed completeness. After de novo assembly, the quality of the assembled genomes (FASTA format) indicated >90% completeness using the BUSCO program. The genome length of *H. burtonii* was 12.3 Mb, with a GC content of 35.16%. The number of contigs in this genome was 191, with L50 and N50 values of 9 and 439,709 bp, respectively (Table 2).

The genomes annotation statistics by QUAST of *H. burtonii* was shown with a total length is 6.8 Mb, GC content is 36.47% and 4,564 contigs. The L50 was 1124 and N50 is 1,875 bp. The high quality of annotated genome allowed the

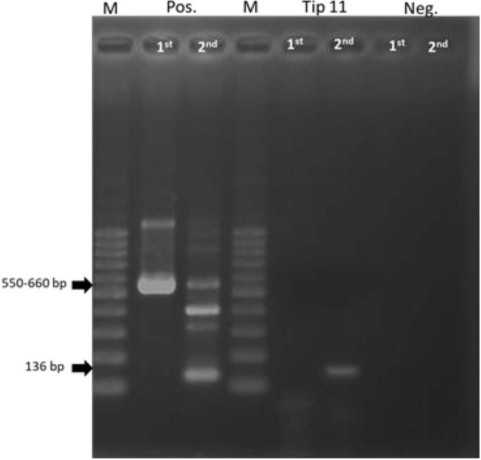

**Fig 2. The result of nested-PCR result of *H. burtonii* (Tip11) comparing with positive control (*Cryptococcus* spp.) and negative control.**

**Table 1. Antifungal susceptibility test results.**

| Name of isolate | Amphotericin B (µg/mL) | | Fluconazole (µg/mL) | | Itraconazole (µg/mL) | | Ketoconazole (µg/mL) | |
|---|---|---|---|---|---|---|---|---|
| | MIC | MFC | MIC | MFC | MIC | MFC | MIC | MFC |
| Tip1 | 0.0625 | 0.5 | 4 | 64 | 0.25 | 4 | 0.0625 | 4 |
| Tip10 | 0.0625 | 0.25 | 8 | 32 | 0.5 | 8 | 0.0625 | 16 |
| Tip11 | 0.25 | 1 | 4 | 64 | >32 | >32 | 32 | >32 |
| Park4 | 0.25 | 0.5 | 0.25 | 32 | 0.125 | 8 | 0.0625 | 4 |

MIC= Minimal Inhibition Concentration, MFC=Minimal Fugicidal Concentration

**Table 2. The quality assessment results of assembled genomes of *H. burtonii* (FASTA format) by QUAST v.5.0.2.**

| Properties | *H. burtonii* TiP11 genome |
| --- | --- |
| Contigs (>= 0 bp) | 191 |
| Contigs (>= 1000 bp) | 76 |
| Total lengh (>= 0 bp) | 12,378,578 |
| Total lengh (>= 1000 bp) | 12,350,388 |
| Genome size (kb) | 12,379 |
| Largest contigs (bp) | 1,210,095 |
| Total length (bp) | 12,360,159 |
| GC content (%) | 35.16 |
| N50 (bp) | 439,709 |
| N90 (bp) | 100,385 |
| auN (bp) | 52,370.8 |
| L50 | 9 |
| L90 | 33 |

prediction of protein-coding genes for *H. burtonii* strain by using OrthoDB database. A total of 2,137 genes and proteins were predicted in the analysis, with most proteins related to yeasts in the *Saccharomycetaceae* family (S1 File). The Complete BUSCOs (C) has 75.5% (n = 1614), 5.2% (n = 111) of Fragmented BUSCOs (F) and 19.3% of Missing BUSCOs (M) remained unannotated (Table 3).

**Table 3. The genomes annotation statistics of *H. burtonii* TiP11 isolate by QUAST v.5.0.2 and BUSCO version of Galaxy version 5.5.0 + galaxy0.**

| Properties | *H. burtonii* TiP11 genome |
| --- | --- |
| Contigs (>= 0 bp) | 4564 |
| Contigs (>= 1000 bp) | 2859 |
| Total length (>= 0 bp) | 6,895,475 |
| Total length (>= 1000 bp) | 5,756,607 |
| Genome size (kb) | 12,379 |
| Largest contigs (bp) | 14,310 |
| Total length (bp) | 6,731,964 |
| GC content (%) | 36.47 |
| N50 (bp) | 1875 |
| N90 (bp) | 885 |
| auN (bp) | 2385.4 |
| L50 | 1124 |
| L90 | 3180 |
| **BUSCO Orthologs (OrthoDB database)** | |
| Genome Completeness (%) | 75.5 |
| Complete and single-copy BUSCOs (S) | 1608 |
| Complete and duplicated BUSCOs (D) | 6 |
| Fragmented BUSCOs | 111 |
| Missing BUSCOs | 412 |
| Total BUSCO groups searched | 2,137 |

The function category has divided to 24 categories including RNA processing and modification, Chromatin structure and dynamics, Energy production and conversion, Cell cycle control, cell division, chromosome partitioning, Amino acid transport and metabolism, Nucleotide transport and metabolism, Carbohydrate transport and metabolism, Coenzyme transport and metabolism, Lipid transport and metabolism, Translation, ribosomal structure and biogenesis, Transcription, Replication, recombination and repair, Cell wall/membrane/envelope biogenesis, Cell motility, Posttranslational modification, protein turnover, chaperones, Inorganic ion transport and metabolism, Secondary metabolites biosynthesis, transport and catabolism, Signal transduction mechanisms, Intracellular trafficking, secretion, and vesicular transport, Defense mechanisms, Extracellular structures, Mobilome: prophages, transposons, Cytoskeleton and Function category not found (Fig 3).

Proteins of *H. burtonii* represented similarity to *Pichiaceae* family protein. Among the identified proteins, midasin, heat shock 70 family proteins, and Csf1 were of particular interest (Fig 4). In the defense mechanisms (V) group, RFT1 was specially considered. Phylogenetic annotated protein sequences and 14 *Hyphopichia* sp. annotated protein sequences showed complete identity to *H. burtonii makgeolli* (Fig 5). Total 43 candidate resistance genes of *H. burtonii* was shown in Fig 6. Briefly, the highest number of resistance genes were found in azole (11) followed by itraconazole (10) vericonazole (8), fluconazole (4), casprofungin (4), micafungin (4), and 5-fluorocystosine (2), respectively. In the Azole group, ATP binding cassette (ABC) transporters superfamily (SNQ2, MDL1, MDL2 and ADP1) was the most finding. While in itraconazole found cytochrome P450 family including ERG11, ERG5, HAP5 and COX10.

## Discussion

The morphology of *H. burtonii* observed in this study was consistent with previous descriptions by Burgain [19]. As previously noted, the cells are short, ellipsoidal, or elongate, with pseudohyphae and true hyphae sometimes forming. Colonies are tannish-white, and their texture ranges from butyrous to hyphal [20]. *H. burtonii*, historically referred to as 'chalky mould', is known to cause food spoilage by fragmenting hyphae into short lengths [21]. Despite its role as a spoilage organism, it has also been used in bread and dairy production and is part of the flora of cured meats. To date, it has only been reported as the cause of cutaneous mycosis in a bat [22].

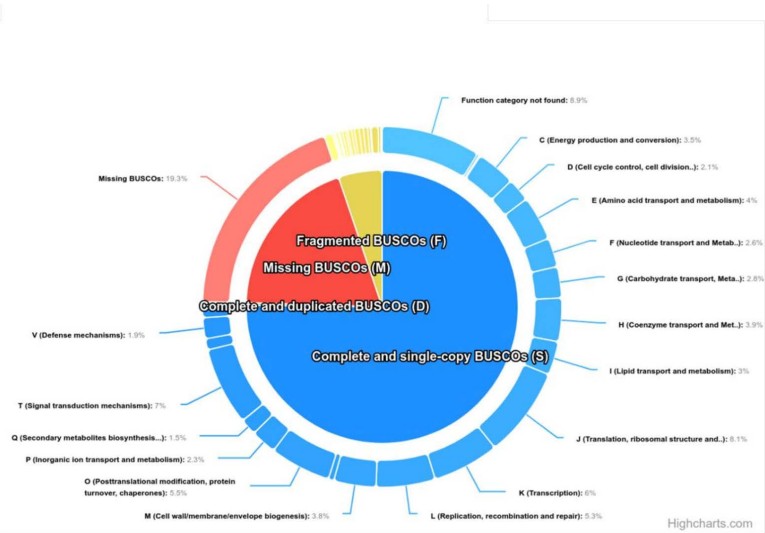

**Fig 3. Representative of Complete BUSCOs (C), Fragmented BUSCOs (F) and Missing BUSCOs (M) proteins of *H. burtonii* annotated sequences and function category has divided to 24 categories.**

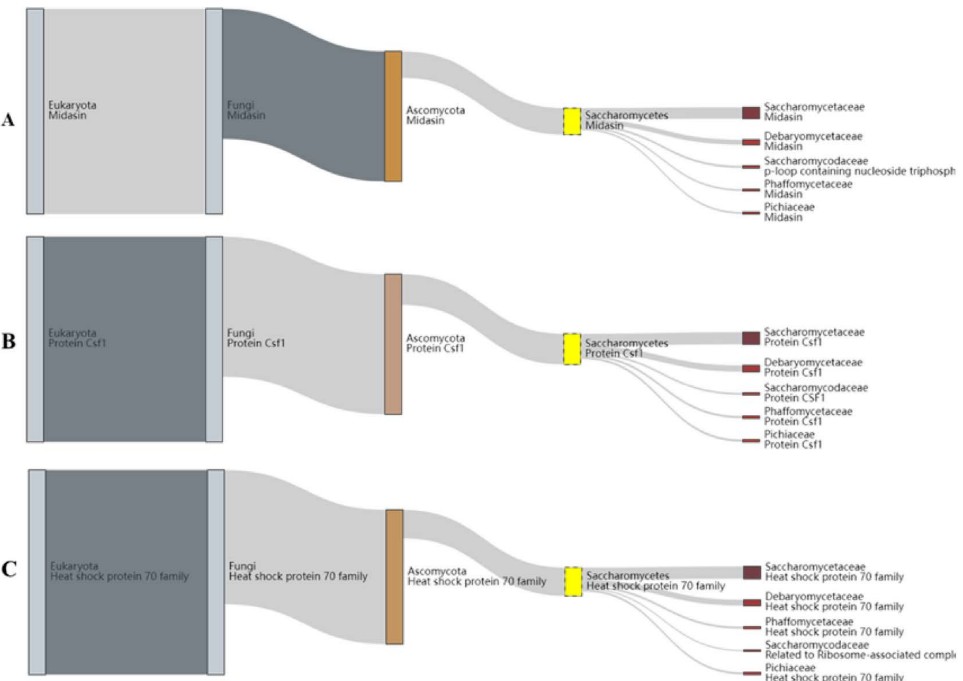

**Fig 4. Representative proteins of *H. burtonii* (A) Midasin (B) Csf1 (C) Heat shock protein 70.**

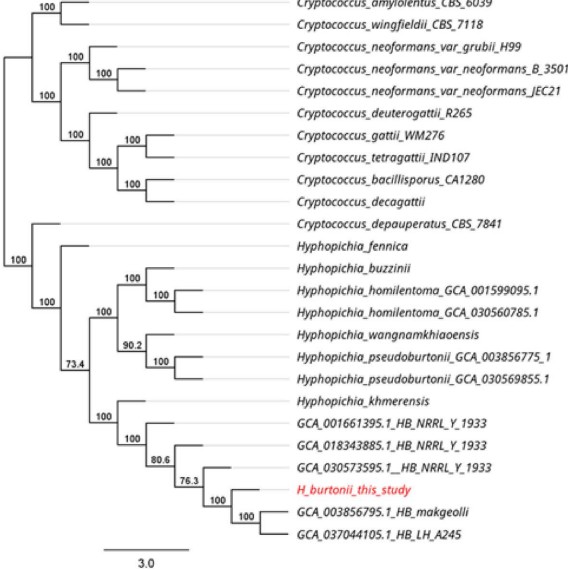

**Fig 5. Phylogenetic of 11 *Cryptococcus* sp. annotated protein sequences and 14 *Hyphopichia* sp. annotated protein sequences.**

The results of the drug susceptibility test showed that *H. burtonii* isolate was highly susceptible to amphotericin B, exhibited intermediate resistance to fluconazole, and was highly resistant to itraconazole and ketoconazole. Relatively with the finding of candidate resistance genes by HDMM analysis (Fig 6), which were high in azole and itraconazole. The

| Drug | Query | Description (gene annotation) | Preferred name | COG category |
|------|-------|------------------------------|----------------|--------------|
| Azoles | 1.g167 | Belongs to the ABC transporter superfamily | - | Q |
| | 49.g4493 | Belongs to the ABC transporter superfamily | - | Q |
| | 23.g3658 | Belongs to the ABC transporter superfamily | - | Q |
| | 49.g4492 | Belongs to the ABC transporter superfamily | SNQ2 | Q |
| | 9.g2289 | ABC-2 type transporter | - | Q |
| | 12.g2797 | ABC transporter transmembrane region | MDL1 | Q |
| | 5.g1509 | ABC transporter transmembrane region | - | Q |
| | 3.g1157 | ABC transporter transmembrane region | MDL2 | Q |
| | 8.g2071 | AAA domain, putative AbiEii toxin, Type IV TA system | ADP1 | Q |
| | 1.g455 | Saccharomyces cerevisiae YMR221c | - | S |
| | 40.g4338 | Vacuolar multidrug resistance ABC transporter | - | Q |
| Fluconazole | 12.g2727 | Belongs to the cytochrome P450 family | ERG11 | Q |
| | 12.g2726 | Belongs to the cytochrome P450 family | ERG11 | Q |
| | 4.g1314 | Histone-like transcription factor (CBF/NF-Y) and archaeal histone | HAP3 | K |
| | 3.g1087 | Transcriptional activator | HAP5 | K |
| Caspofungin | 25.g3801 | synthase | GSL2 | M |
| | 24.g3758 | synthase | - | M |
| | 18.g3365 | 1,3-beta-glucan synthase subunit FKS1 | - | M |
| | 15.g3048 | Converts protoheme IX and farnesyl diphosphate to heme O | COX10 | H |
| Micafungin | 25.g3801 | synthase | GSL2 | M |
| | 24.g3758 | synthase | - | M |
| | 18.g3365 | 1,3-beta-glucan synthase subunit FKS1 | - | M |
| | 12.g2710 | Saccharomyces cerevisiae YDL012C | - | S |

| Drug | Query | Description (gene annotation) | Preferred name | COG category |
|------|-------|------------------------------|----------------|--------------|
| Itraconazole | 12.g2726 | Belongs to the cytochrome P450 family | ERG11 | Q |
| | 12.g2727 | Belongs to the cytochrome P450 family | ERG11 | Q |
| | 3.g1087 | Transcriptional activator | HAP5 | K |
| | 2.g760 | cytochrome P450 | - | Q |
| | 6.g1677 | Belongs to the cytochrome P450 family | ERG5 | Q |
| | 4.g1189 | Belongs to the cytochrome P450 family | - | Q |
| | 15.g3048 | Converts protoheme IX and farnesyl diphosphate to heme O | COX10 | H |
| | 2.g720 | Belongs to the cytochrome P450 family | - | Q |
| | 15.g3081 | cytochrome P450 | - | Q |
| | 14.g3022 | Cytochrome P450 | - | Q |
| Voriconazole | 12.g2726 | Belongs to the cytochrome P450 family | ERG11 | Q |
| | 12.g2727 | Belongs to the cytochrome P450 family | ERG11 | Q |
| | 6.g1677 | Belongs to the cytochrome P450 family | ERG5 | Q |
| | 2.g760 | cytochrome P450 | - | Q |
| | 4.g1189 | Belongs to the cytochrome P450 family | - | Q |
| | 14.g3022 | Cytochrome P450 | - | Q |
| | 2.g720 | Belongs to the cytochrome P450 family | - | Q |
| | 15.g3081 | cytochrome P450 | - | Q |
| 5-fluorocytosine | 12.g2700 | Uracil phosphoribosyltransferase | FUR1 | TZ |
| | 1.g134 | cytosine deaminase | FCY1 | F |

**Fig 6. List of candidate antifungal resistance genes found after HMM Analysis using the ResFungi HMM Databasea in *H. burtonii* Tip11 isolates.**

most finding resistance genes in azole group were ABC transporters superfamily that are responsible for drug resistance and a low bioavailability of drugs by pumping a variety of drugs out cells at the expense of ATP hydrolysis [23]. ATP-binding cassette (ABC) transporters play a role in active efflux of a broad range of xenobiotics which is one of the common mechanisms of multidrug resistance in eukaryotic cells [24]. Azole antifungal drugs inhibit cytochrome P450-dependent enzymes, particularly 14-demethylase, which is crucial for the biosynthesis of ergosterol—a key component of fungal cell membrane structure and function [25]. Resistance to azoles often involves alterations in genes encoding the azole drug target 14-α-demethylase (ERG11) in yeast [26]. ERG11 gene was identified in the *H. burtonii* isolate possess ergosterol biosynthesis genes might contribute to its resistance to ketoconazole.

These findings differ significantly from reports of *H. burtonii* causing sterile peritonitis in humans, where the yeast was highly susceptible to amphotericin B, voriconazole, fluconazole, itraconazole, and caspofungin [27]. The observed MIC levels of ketoconazole in this yeast are like those reported for *C. neoformans* var. neoformans and *C. neoformans* var. gattii (16–64 µg/mL) [28]. However, this finding showed similar with the recent study that reported *C. gattii* VGIII was less susceptible to fluconazole and itraconazole [29].

The genome of this *H. burtonii* isolate contains the Pichiaceae family protein midasin (Fig 4A), which is involved in coenzyme transport and metabolism, intracellular trafficking, secretion, and vesicular transport. Previous studies have suggested that midasin may function as a nuclear chaperone and be involved in the assembly/disassembly of macromolecular complexes within the nucleus of yeast [30]. Interestingly, *H. burtonii* also possesses the Csf1 gene (Fig 4B), which plays a critical role in protein uptake at low temperatures, like the Hsf1 protein in Saccharomycetes [31]. Additionally, the heat shock protein 70 family (Fig 4C) was identified in this yeast, which is known to participate in both the heat shock response and adaptation to oxidative stress and glucose starvation [32]. These characteristics may explain why *H. burtonii* is a yeast species commonly found in nature and highly competitive under harsh environmental conditions, such as low water availability (0.85 aw) [19] and exhibited strong exhibit strong halotolerance [33]. Moreover, it is a potent producer of amylases that support mite digestion in stored products such as cereals and grains [34]. Interestingly, RFT1 gene in categories of defense mechanisms was identified in *H. burtonii* genome. RFT1 catalyzes the translocation of M5GN2-PP-Dol across the lipid bilayer and acts as the M5GN2-PP-Dol ER flippase [35]. Yeast RFT1 is homologous to human RFT1 and SEC61A1 and has been used to study congenital disorder of glycosylation type I and cancer [36].

However, infections in animals and humans caused by *H. burtonii* are rarely reported. There have been cases in which *H. burtonii* caused dermatitis in Barbastella bats [22]. The yeast has also been isolated from fish feed (tambatinga fish) and marine environments, including fish and fishponds [37,38]. The presence of this yeast may depend on the composition of the feed [38]. Laying hens might harbour potentially pathogenic *H. burtonii* yeasts in their gastrointestinal tracts and spread them through faeces and eggs [8]. Human cases of secondary peritonitis involving *H. burtonii* have also been documented [27,39].

## Conclusion

This study is the first report of *H. burtonii* recovered form zebra dove droppings in Thailand with the novel knowledge on the resistance genes. The azole drugs (Itraconazole and ketoconazole) resistance were observed in this yeast suggest that infections originating from zebra dove droppings could pose significant public health risks, especially in immunocompromised individuals, like other pathogenic yeasts where treatment options are limited. Therefore, further characterization and functional studies of *H. burtonii* are necessary. Additionally, implementing biosecurity measures in zebra dove farms is crucial to mitigate potential risks.

## Supporting information

**S1 File. The lineage dataset of proteins in *H. burtonii*.** The most proteins of *H. burtonii* related to yeasts in the *Saccharomycetaceae* family

(CSV)

AcknowledgmentThis research sincerely thank to the Faculty of Veterinary Science, Prince of Songkla University supported for the equipments. We are grateful to the zebra dove farmers who granted us permission to enter their farms and collect samples.

## Author contributions

**Conceptualization:** Saowakon Indoung.

**Data curation:** Ruttayaporn Ngasaman, Sanicha Chumtong, Komwit Surachat, Rattanaruji Pomwised.

**Formal analysis:** Saowakon Indoung, Sakaoporn Prachantasena.

**Funding acquisition:** Ruttayaporn Ngasaman.

**Investigation:** Sanicha Chumtong, Ratchakul Wiriyaprom.

**Project administration:** Ratchakul Wiriyaprom.

**Resources:** Sakaoporn Prachantasena.

**Software:** Komwit Surachat.

**Supervision:** Rattanaruji Pomwised.

**Validation:** Ruttayaporn Ngasaman.

**Visualization:** Ruttayaporn Ngasaman.

**Writing – original draft:** Sanicha Chumtong.

**Writing – review & editing:** Ruttayaporn Ngasaman.

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
