## [Decision Letter · Decision Letter 0]

13 Feb 2025

Dear Dr. Ngasaman,

Thank you for submitting your manuscript to PLOS ONE. After careful consideration, we feel that it has merit but does not fully meet PLOS ONE’s publication criteria as it currently stands. Therefore, we invite you to submit a revised version of the manuscript that addresses the points raised during the review process.

We look forward to receiving your revised manuscript.

Kind regards,

Abhay K. Pandey

Academic Editor

PLOS ONE

Journal Requirements:

https://www.mdpi.com/2309-608X/8/3/246

In your revision ensure you cite all your sources (including your own works), and quote or rephrase any duplicated text outside the methods section. Further consideration is dependent on these concerns being addressed.

Funding from Faculty of Veterinary Science, Prince of Songkla University

Grant no. VET6604128S

This research was supported by a research grant (Grant no. VET6604128S) from the Faculty of Veterinary Science, Prince of Songkla University. We are grateful to the zebra dove farmers who granted us permission to enter their farms and collect samples.

Funding from Faculty of Veterinary Science, Prince of Songkla University

Grant no. VET6604128S

5. Please provide a complete Data Availability Statement in the submission form, ensuring you include all necessary access information or a reason for why you are unable to make your data freely accessible. If your research concerns only data provided within your submission, please write "All data are in the manuscript and/or supporting information files" as your Data Availability Statement.

6. Please include a copy of Table 1 which you refer to in your text on page 7. 

Reviewers' comments:

Reviewer's Responses to Questions

**Comments to the Author**

1. Is the manuscript technically sound, and do the data support the conclusions?

Reviewer #1: Yes

Reviewer #2: Yes

2. Has the statistical analysis been performed appropriately and rigorously?

Reviewer #1: N/A

Reviewer #2: N/A

3. Have the authors made all data underlying the findings in their manuscript fully available?

Reviewer #1: Yes

Reviewer #2: Yes

4. Is the manuscript presented in an intelligible fashion and written in standard English?

Reviewer #1: Yes

Reviewer #2: Yes

Reviewer #1: L 26 (Geopelia striata): should be italic

L28 Cryptococcus spp.: “Cryptococcus” should be italic

All species should be italicized through the manuscript

L27-28” Four contaminated isolates were found with morphology and biochemical test results like those of Cryptococcus spp.,” please write percentage of positive samples from total examined samples.

L31-32 “The resistance isolate (Tip11) was subjected to whole genome sequencing.”: should be omitted as it was mentioned at L34-36

L43-44 “Moreover, further characterisation and functional analysis of the proteins identified should be considered.”: what is the need of this sentence??

L51-61: a reference is required

L63 “Trichomonas gallinae” is a parasite so it should be omitted.

L72-73 “Examples of zoonotic yeasts associated with bird droppings include Histoplasma capsulatum, Aspergillus niger,”: Authors should delete Histoplasma capsulatum, Aspergillus niger as they mentioned examples for yeasts. Histoplasma capsulatum is a dimorphic fungus and Aspergillus niger is mould.

L87-88 “0.5 McFarland (1–5 × 106 CFU/mL) using a spectrophotometer (absorbance = 530 nm) 88 and diluted to 0.5–2.5 × 103 CFU/mL”: please adjust concentrations.

L116 “CLSI” write in full at the first mention

L120-121 “The antifungal drugs amphotericin B, fluconazole, 121 ketoconazole, and itraconazole” please write the source of each drug and mention the solvent.

L130 “ MIC endpoints were read according to NCCLS criteria [13].” The authors mentioned at L116 “CLSI document M27-A3 [10]” why they read according to NCCLS criteria?

L146-147” four contaminated isolates, initially presumed to be Cryptococcus spp. based on their morphology and the results of biochemical tests”: please mention the biochemical characters.

Line 180-181: “The function category has divided to 24 categories including” please write categories without letters (A, B,…………)

Authors should write resistance genes that were found in WGS as the isolate was resistant. Moreover, virulence genes should be mentioned.

In figure 1:budding yeast cells with Gram stain instead of Crystal violet.

In figure 2 please write the positive control in figure legend. Where is the negative control?

L 203- 204 “The results of this study showed that H. burtonii isolate was highly susceptible to amphotericin B, exhibited intermediate resistance to fluconazole, and was highly resistant to itraconazole and ketoconazole. “:)” this finding should be supported with this recent study https://doi.org/10.1186/s43008-024-00153-w that reported “C. gattii VGIII was less susceptible to fuconazole and itraconazole”

L213- 214 “The observed MIC levels of ketoconazole in this yeast are like those reported for C. neoformans var. neoformans and C. neoformans var. gattii (16–64 µg/mL)” this finding should be supported also with this recent study https://doi.org/10.1186/s43008-024-00153-w

Reviewer #2: 1. It needs to add the importance of this agent as zoonotic in the introduction.

2. Needs to explain about the novelty.

3. Edit and check all MS for format, such as line 28 cryotococcus should be italic.

**Do you want your identity to be public for this peer review?** For information about this choice, including consent withdrawal, please see our Privacy Policy

Reviewer #1: No

Reviewer #2: **Yes: ** Abdollah Derakhshandeh

---

## [Author Response · Author response to Decision Letter 1]

12 Mar 2025

They gave all the benefit comments to our research article and would like to thank all of the reviewers.

---

## [Decision Letter · Decision Letter 1]

28 May 2025

Whole-genome sequencing of Hyphopichia burtonii from isolated yeast recovered from zebra dove droppings in Thailand

PONE-D-24-53241R1

Dear Dr.Ngasaman,

We’re pleased to inform you that your manuscript has been judged scientifically suitable for publication and will be formally accepted for publication once it meets all outstanding technical requirements.

Kind regards,

Hamida Hamdi Mohammed Ismail, ph.D.

Academic Editor

PLOS ONE

Additional Editor Comments (optional):

Reviewers' comments:

Reviewer's Responses to Questions

**Comments to the Author**

Reviewer #1: All comments have been addressed

Reviewer #2: All comments have been addressed

2. Is the manuscript technically sound, and do the data support the conclusions?

Reviewer #1: Yes

Reviewer #2: Yes

3. Has the statistical analysis been performed appropriately and rigorously?

Reviewer #1: Yes

Reviewer #2: N/A

4. Have the authors made all data underlying the findings in their manuscript fully available?

Reviewer #1: Yes

Reviewer #2: Yes

5. Is the manuscript presented in an intelligible fashion and written in standard English?

Reviewer #1: Yes

Reviewer #2: Yes

Reviewer #1: The authors have addressed all comments and the manuscript is suitable for publication. There are no other comments to authors.

Reviewer #2: I have checked my raised questions. I think the authors tried to address all issues. So, I do not have more commensts.

**Do you want your identity to be public for this peer review?** For information about this choice, including consent withdrawal, please see our Privacy Policy

Reviewer #1: **Yes: ** Yasmine Hasanine Tartor

Reviewer #2: **Yes: ** Abdollah Derakhshandeh

---

## [Editor Report · Acceptance letter]

PONE-D-24-53241R1

PLOS ONE

Dear Dr. Ngasaman,

I'm pleased to inform you that your manuscript has been deemed suitable for publication in PLOS ONE. Congratulations! Your manuscript is now being handed over to our production team.

Kind regards,

on behalf of

Professor Hamida Hamdi Mohammed Ismail

Academic Editor

PLOS ONE